# Ependymal cilia beating induces an actin network to protect centrioles against shear stress

Alexia Mahuzier[1], Asm Shihavuddin[1,6], Clémence Fournier[2,7], Pauline Lansade[1], Marion Faucourt[1], Nikita Menezes[1], Alice Meunier[1], Meriem Garfa-Traoré[3], Marie-France Carlier[2], Raphael Voituriez[4,5], Auguste Genovesio [1], Nathalie Spassky[1] & Nathalie Delgehyr[1]

Multiciliated ependymal cells line all brain cavities. The beating of their motile cilia contributes to the flow of cerebrospinal fluid, which is required for brain homoeostasis and functions. Motile cilia, nucleated from centrioles, persist once formed and withstand the forces produced by the external fluid flow and by their own cilia beating. Here, we show that a dense actin network around the centrioles is induced by cilia beating, as shown by the disorganisation of the actin network upon impairment of cilia motility. Moreover, disruption of the actin network, or specifically of the apical actin network, causes motile cilia and their centrioles to detach from the apical surface of ependymal cell. In conclusion, cilia beating controls the apical actin network around centrioles; the mechanical resistance of this actin network contributes, in turn, to centriole stability.

[1] Cilia biology and neurogenesis, Institut de biologie de l'Ecole normale supérieure (IBENS), Ecole normale supérieure, CNRS, INSERM, PSL Université Paris, 75005 Paris, France. [2] Biochimie Biophysique et Biologie Structurale, I2BC, CNRS, 91198 Gif-sur-Yvette, France. [3] Cell Imaging Platform, INSERM US24 Structure Fédérative de Recherche Necker, Paris Descartes Sorbonne Paris Cité University, Paris, France. [4] Laboratoire de Physique Théorique de la Matière Condensée, CNRS/UPMC, Paris, France. [5] Laboratoire Jean Perrin, CNRS/UPMC, Paris, France. [6]Present address: Department of Applied Mathematics and Computer Science, Technical University of Denmark, Kgs, Lyngby, Denmark. [7]Present address: UPMC University Paris 06 UM 1127, Inserm U1127, CNRS UMR7225, ICM, Sorbonne Université, Paris, France. These authors contributed equally: Nathalie Spassky, Nathalie Delgehyr. Correspondence and requests for materials should be addressed to N.S. (email: spassky@biologie.ens.fr) or to N.D. (email: delgehyr@biologie.ens.fr)

Multiciliated cells ensure the displacement of liquid or mucus, which perform essential functions in the organism, such as the displacement of oocytes in fallopian tubes, the clearance of mucus from the airways, the stirring of luminal fluid in the efferent ducts or the circulation of cerebrospinal fluid (CSF) in brain ventricles[1,2]. Multiciliated cells in the brain, called ependymal cells, line all brain ventricles and form a protective barrier[3]. They also contribute to the neural stem cell niche[4]. Ependymal cilia beating ensures the CSF flow necessary for brain homoeostasis, toxin washout, delivery of signalling molecules and orientation of the migration of newborn neurons[5]. Defective cilia motility is associated with hydrocephalus, which increases pressure in the skull due to an increase in CSF in the ventricular cavities[6]. Ependymal cells are generated from radial glial cells during early postnatal stages[7]. Their regeneration during aging or under pathological conditions is limited[8], resulting in the partial loss of protection of the brain parenchyma in aged patients[9,10]. Most ependymal cells persist, however, throughout life.

Cilia beating results in mechanical constraints on the cells. For example, in the multiciliated organism *Tetrahymena thermophila*, if the ultrastructure of centrioles is abnormal, the forces exerted at the base of beating cilia lead to centriole disassembly[11,12]. To determine how centrioles withstand these mechanical constraints lifelong, we focussed on the potential role of the actin cytoskeleton[13]. The actin cytoskeleton of multiciliated cells has been the focus of detailed analysis. Notably, in the multiciliated epidermal cells of *Xenopus* larvae, it was demonstrated that two interconnected apical and subapical actin networks form a 3D-network that connects the centrioles, thus contributing to their spacing and to the synchronisation of cilia beating[14]. The actin cytoskeleton is assembled and organised progressively at the apical surface of multiciliated progenitor cells[15–19]. It contributes to the intercalation of the multiciliated cells in the *Xenopus* epidermis[17,18], guides the apical migration and docking of newly formed centrioles in the plasma membrane[14–16,20,21] and participates to ciliogenesis[16]. However, how this actin cytoskeleton network is assembled in ependymal cells and what its direct links with centriole stability are have not been addressed so far. In this study, we describe the coordination between ciliary beating and actin organisation that actively contributes to the protection of ependymal cilia and centrioles against the shear stress generated by ciliary beating and the associated fluid flow.

## Results

### Actin assembles around centrioles during development

To investigate when and how the actin network develops around ependymal cell centrioles, lateral ventricular walls were immunostained with antibodies against cilia, centrioles and filamentous actin (F-actin) at different stages during motile cilia formation. The same area was analysed throughout the paper (Fig. 1a). F-actin is localised at cell borders at all stages (Fig. 1a). In cells in which centrioles are not yet ciliated (post-natal day 4, P4), actin staining at the apical surface (defined by the localisation of the distal centriolar marker FOP[22]) is scarce and diffuse. The staining intensifies all over the cell cortex as ciliation begins (P6). As motile cilia elongate during cell maturation, actin accumulates in the centriolar patch (Fig. 1a, b), and is positioned asymmetrically at the front of mature ependymal cells[23] (P15). At this stage, the actin network is thicker, extending from the apical surface of the cells (apical actin) to ~1 μm below (subapical actin). Note that at P15, the apical actin network consists of thick actin cables oriented towards the rear of the cell, whereas the subapical network contains smaller dot-like actin-positive structures in the centriolar patch (P15, Fig. 1a).

To characterise the morphology of the apical and subapical actin networks connected to centrioles in mature ependymal cells (P15), we used high resolution microscopy (STED, Fig. 1c, d) and 3D-modelling (Fig. 1e). Co-staining of actin with markers of the distal appendages (Cep164[24]) or the distal lumen of the centrioles (centrin[25]) revealed that the apical actin network is distributed around the centrioles and extends from the distal appendages to the centrin-positive distal portion of the centrioles (Fig. 1c, f). We occasionally observed apical actin filaments surrounding clusters of centrioles, the distal appendages of which were merged (Fig. 1d). Ependymal centrioles or clusters of centrioles are anchored in the ciliary pocket, a depression of the plasma membrane[26], suggesting that the apical actin filaments are organised at this level. Co-staining of actin with markers of the proximal end of the centrioles (Akap450[27]) and rootlets (Rootletin[28]) shows that the subapical actin network extends from the proximal end of the centrioles to their rootlets (Fig. 1c, f). The apical and subapical actin networks overlap partially along the Z-axis and link together individual or clusters of centrioles (Fig. 1e). Therefore, in multiciliated ependymal cells, centrioles are embedded in a three-dimensional actin structure from their distal appendages to their rootlets. The assembly and organisation of this complex actin network in the centriolar patch correlates with the elongation of motile cilia (Fig. 1a, b). It is unclear, however, whether ciliogenesis induces the assembly of the actin network at the centriolar patch or, conversely, whether the increase in actin assembly leads to cilia elongation.

### Cilia beating induces apical actin assembly at centrioles

To assess the role of cilia in actin organisation, we studied mice in which genes essential for the development of cilia, such as *Kif3A*[29] (a subunit of the motor protein kinesin-2) or *Ift88*[30] (a homologue intraflagellar transport protein 88) or for cilia motility, such as *hydin*[31] (that causes defects in the central pair of motile cilia when mutated) or Lrrc6 (axonemal dynein subunit[32–34]) were mutated or depleted.

We used *hGFAP-Cre; Kif3A* conditional knocked out mutant mice (*Kif3A*-cKO) that have normal primary cilia in ependymal cell progenitors, but are virtually devoid of motile cilia on the lateral wall of the lateral ventricles[35] (Supplementary Figure 1a). F-actin fluorescence immunostaining showed that actin was distributed homogeneously on the apical cell surface in mature *Kif3A*-cKO ependymal cells, whereas an additional apical actin network was observed in the centriolar patch in controls (Fig. 2a, b). In the *Ift88* conditional mutant (*Ift88*-cKO), similar results were obtained (Supplementary Figure 1b). Apical actin enrichment at the centriolar patch correlates with cilia elongation (Fig. 1a, b) and two independent mutant mice in which motile cilia are absent, lack the apical centriolar actin network (Fig. 2a, b; Supplementary Figure 1b). Thus, these results indicate that motile cilia are a pre-requisite for actin enrichment at the centriolar patch.

Unlike the apical actin network, the subapical centriolar distribution of actin was identical in the wild-types and the ciliary mutants (Fig. 2a, c; Supplementary Figure 1b). Therefore, the organisation of the subapical actin network around centrioles is independent of both motile cilia and enrichment of the apical actin network in the centriolar patch. Since both ciliary mutant mice (*Kif3A*-cKO and *Ift88*-cKO) are devoid of motile cilia, it is unclear whether the cilia themselves or their motility is required for assembly of apical actin around the centrioles. To determine whether cilia motility is required, we used *hydin* mutant mice that have stiff vibrating cilia[31]. Immunostaining of the cilia in *hydin* mutant mice confirmed that mature (P15) ependymal cells extend cilia that are normal in terms of length and density

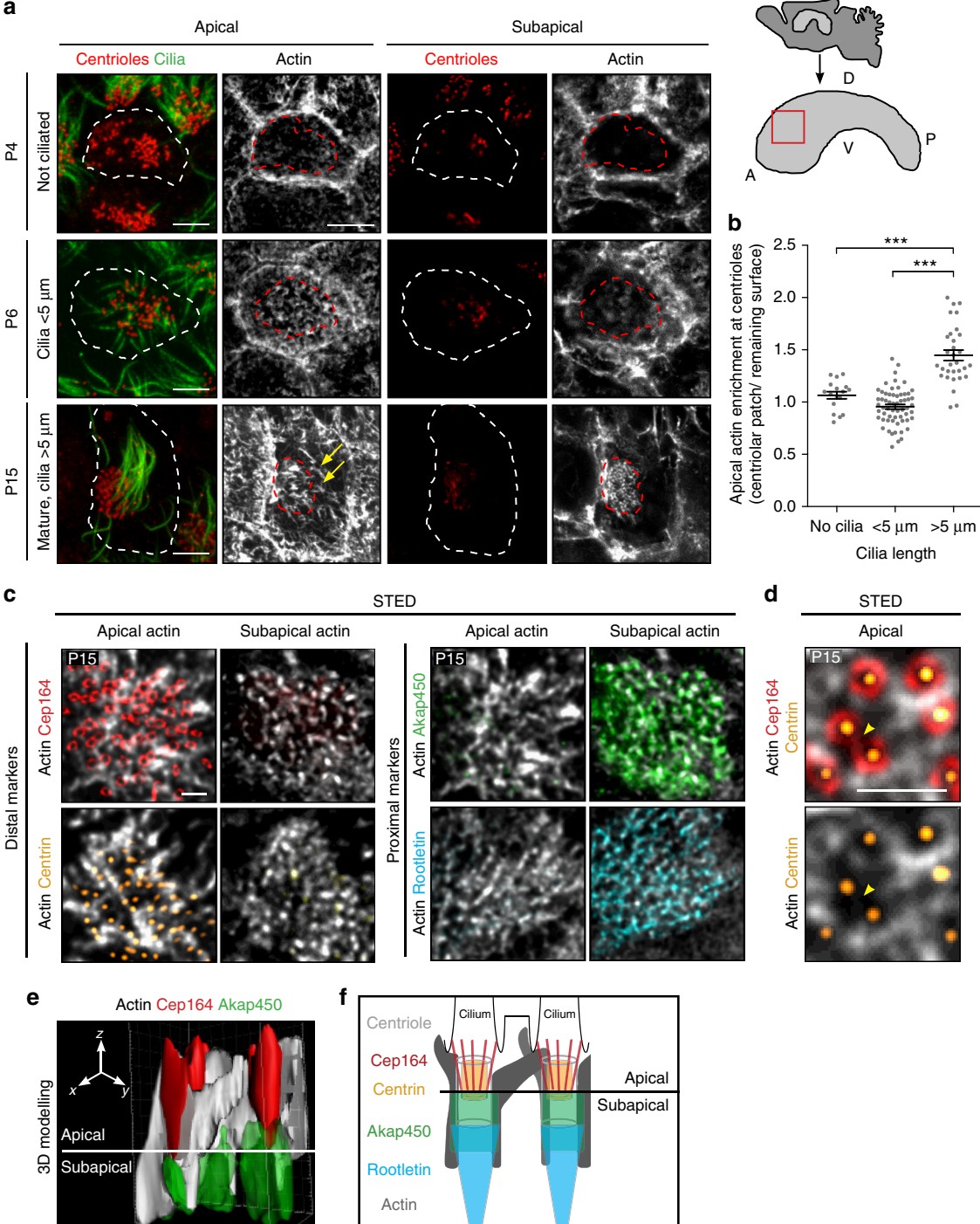

**Fig. 1** Formation of a dense actin network at the centriolar patch during ependymal cell differentiation **a** F-actin (phalloidin, grey), centrioles (FOP, red) and cilia (GT335, green) in whole mounts of lateral ventricular walls at postnatal days 4 (P4), P6 and P15 at the apical level (colocalising with FOP staining) and subapical level; the actin network in the centriolar patch thickens as motile cilia grow; cell borders and centriolar patches are outlined with dashed white and red lines, respectively. Arrows point to long actin filaments. On the right, a sagittal view of the lateral ventricular wall and the region of interest analysed throughout the paper (red square); A: Anterior, P: Posterior, D: Dorsal, V: Ventral. **b** Ratio of mean phalloidin fluorescence intensities in the centriolar patch (red dashed lines in **a**) and on the remaining cell surface (white minus red dashed lines) as a function of ciliary length. Error bars represent the sem of $n = 112$ cells from three mice; $P$-values were determined by one-way ANOVA followed by Dunn's multiple comparison test; ***$P \leq 0.0001$. **c–e** High resolution microscopy (STED) of mature ependymal cells on P15 whole-mount lateral ventricular walls. **c** Staining of F-actin (phalloidin, grey) and centriole distal markers, Cep164 (distal appendages, red) or Centrin2-GFP (distal lumen, yellow) that colocalise with the apical actin network, or centriole proximal markers Akap450 (pericentriolar material, green) or Rootletin (rootlets, cyan) that colocalise with the subapical actin network. **d** Zoom-in of Cep164 (red)-, Centrin2-GFP (yellow)- and F-actin (phalloidin, grey)-labelled mature ependymal cells; note a cluster of two centrioles with merged distal appendages (arrowhead) surrounded by actin. **e** Three-dimensional modelling of Cep164 (red)-, Akap450 (green)- and F-actin-(grey)-positive centrioles showing the links between the apical and subapical networks, and between neighbouring centrioles. **f** Model of centrioles with their associated actin bundles positioned according to the markers used in this study. Scale bars=5 μm in **a** and 1 μm in **c**, **d**

(Supplementary Figure 1a, c). Line scans (kymograms) and quantification of cilia beating frequency on side views of videos of motile cilia showed that the beating frequency was reduced in *hydin* mutant mice (Fig. 2d) and its amplitude abnormal (Supplementary Figure 1d), confirming that *hydin* mutant cilia are stiff and beat slowly[31]. F-actin immunostaining in mature ependymal cells in *hydin* mutants showed that the subapical actin network organisation was unchanged compared to control cells (Fig. 2a, c), but the apical actin enrichment in the centriolar patch

was decreased (Fig. 2a, b), suggesting that cilia motility is required for the apical enrichment of actin around the centrioles of mature ependymal cells.

To confirm whether cilia motility is required for the enrichment in apical actin at the centriolar patch, we blocked cilia motility by using an shRNA against Lrrc6, a subunit of axonemal dynein required for this motility[32–34]. We validated this strategy by the observation of a decrease in the frequency of cilia beating with high speed video microscopy on Lrrc6-depleted

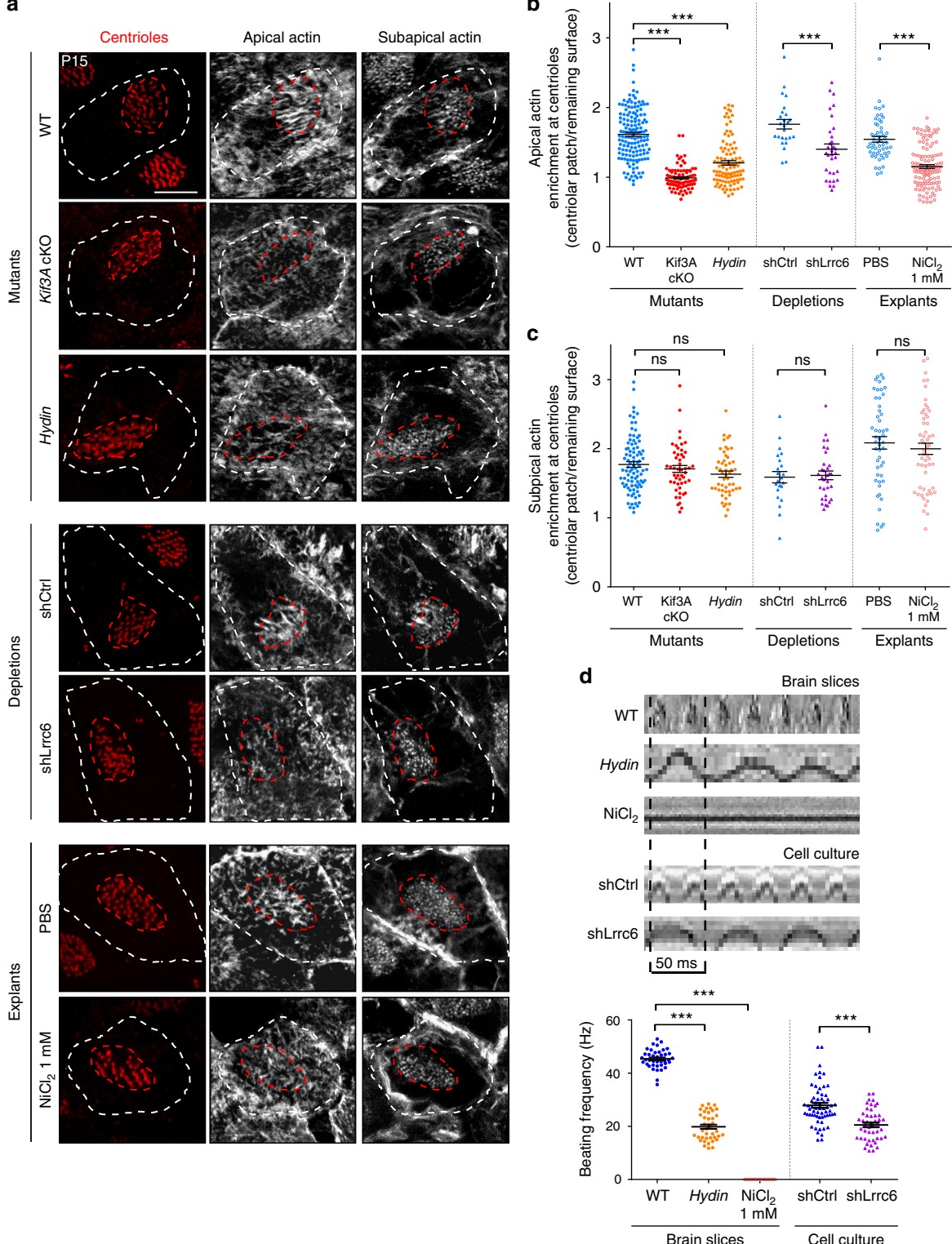

ependymal cells in culture[36,37] (Fig. 2d). Electroporation of the shRNA construct in the lateral ventricles of new-born mice[37,38] and immunostaining of cilia at P15 showed that Lrrc6 depletion did not affect cilia length and density (Supplementary Figure 1a, c). Analysis of F-actin immunostaining at P15 revealed that, in absence of Lrrc6, apical actin was not enriched at the centriolar patch whereas the organisation of subapical actin remained unaffected (Fig. 2a–c). We also measured F-actin fluorescence intensity at the apical and subapical levels compared to surrounding none-transfected cells, and showed that both apical and subapical actin levels are decreased in Lrrc6-depleted mature ependymal cells (Supplementary Figure 1e, f). Altogether, our results suggest that cilia motility is required for the enrichment of apical actin at the centriolar patch and for the assembly of a subpopulation of actin cables at the apical and subapical levels.

To assess whether cilia beating is also required to actively maintain the apical actin enrichment at the centriolar patch, we blocked cilia beating in explants of the lateral ventricular wall of P15 mice, which are lined with ependymal cells bearing motile cilia and fully developed actin networks. To block cilia beating, we used NiCl$_2$, which has a direct effect on axonemal dynein[11,12,39]. Video-recording of treated cilia confirmed that the cilia stopped beating in the treated cells (Fig. 2d; Supplementary Figure 1d), whereas cilia length was unaffected (Supplementary Figure 1a, c). F-actin immunostaining showed, however, a significant decrease in apical but not subapical actin enrichment at the centriolar patch (relative to control explants treated with PBS), when cilia motility was stopped (Fig. 2a–c).

Thus, using genetics or pharmacological treatments, we show that cilia motility is required to initiate and maintain the apical actin enrichment at the centriolar patch.

**Actin maintains centrioles in mature ependymal cells.** To assess the role in centriolar function of actin assembled around centrioles, we treated P15 explants of lateral ventricular wall, lined with ependymal cells bearing motile cilia and assembled actin networks, with cytochalasin-D, which progressively depolymerised the actin network at both the apical and subapical levels and at cell borders, in a dose- and time-dependent manner (Fig. 3a). Neither the length of the motile cilia (Supplementary Figure 2a, b) nor their beating frequency (Fig. 3b) was significantly affected, even by a long treatment with a high concentration (2 µM) of cytochalasin-D. However, cells, cilia tufts and centriolar patches were smaller after cytochalasin-D treatment (Fig. 3a; Supplementary Figure 2a, c) and the intercentriolar distance was slightly decreased (Supplementary Figure 2d), as previously described in multiciliated *Xenopus* cells[14]. In addition, the number of centrioles also decreased significantly in cytochalasin-D-treated explants (Fig. 3c, d). This decrease was time- and dose-dependent (Fig. 3c, e, f). The decrease in the size

of the centriolar patch observed after actin disruption (Supplementary Figure 2c) was thus due to both the loss of centrioles and the decrease in the inter-centriolar distance. Altogether, actin contributes to centriole maintenance and spacing in ependymal cells.

**The apical actin contributes to centriole maintenance.** To determine which actin network, apical or subapical, is required for centriole maintenance, we sought of actin nucleators specifically involved in these networks. There are three different families of actin nucleators, the Arp2/3 complex that nucleates branched actin filaments and the formins and WH2 (Wiskott Aldrich syndrome protein homology 2)-containing proteins that both nucleate linear actin filaments[40]. We treated explants of lateral walls of P15 animals, lined with ependymal cells bearing motile cilia and assembled actin networks, with specific inhibitors of the Arp2/3 complex (CK666[41]) or formins (SMIFH2[42]) to assess their specific contribution to the actin structures of mature ependymal cells. Since we did not observe obvious effect on actin organisation after CK666 treatment (12 h at 100µM, Fig. 2e), we focused on formins. With commercially available antibodies, we found several formins, namely Fhod1, Fmn1, Daam1 and mDia1 that co-localised with centriole markers in ependymal cells (Fig. 4a), as previously shown in multiciliated *Xenopus* epidermal cells for Daam1 and Fmn1[17,43]. Interestingly, compared to cytochalasin-D treatment that affected both the apical and subapical actin network (Fig. 3a), SMIFH2-treatment led to a strong decrease of the apical actin enrichment in the centriolar patch with no major effect on the organisation of the subapical actin network (Fig. 4b–d). These results suggest that formins contribute at least to the maintenance of the apical actin enrichment at the centriolar patch. Interestingly, unlike the effect of cytochalasin-D treatment, no difference in the inter-centriolar distance was observed after SMIFH2-treatment (Supplementary Figure 2f), suggesting that the distance between centrioles is, as previously suggested[14], mainly regulated by the subapical actin organisation (compare Supplementary Figs. 2d and 2f). Formin inhibition as well as actin depolymerisation with cytochalasin-D did not affect cilia length or beating (Supplementary Figure 2a, b; Fig. 4e). Noteworthy, SMIFH2-treatment induced smaller centriolar patches (Supplementary Figure 2g) with decreased number of centrioles relative to DMSO treated explants (Fig. 4f, g). Altogether, these results suggest that, in mature ependymal cells, apical actin in the centriolar patch helps maintain the correct number of centrioles.

To confirm that the apical actin network is implicated in centriole maintenance, we examined the role of the actin binding protein Cordon-Bleu (Cobl), another actin regulator involved in the establishment of apical actin structures such as microvilli[44–46]. Cobl nucleates and severs linear actin filaments in vitro and is

**Fig. 2** Cilia beating is required for the formation and maintenance of the apical actin network at the centriolar patch **a** F-actin (phalloidin, grey) and centrioles (FOP, red) in mature ependymal cells (P15) in ciliary mutant mice (*Kif3A* cKO or *hydin*) and wild-type (WT) mice transfected with shRNA (shCtrl, Control or shLrrc6 directed against the axonemal dynein subunit Lrrc6) or explants of WT ventricular walls treated for 12 h with PBS or 1 mM NiCl$_2$ to block cilia beating; cell borders and centriolar patches are outlined with dashed white and red lines, respectively. Apical actin enriched in the centriolar patch is reduced in ciliary mutants and after NiCl$_2$ treatment, whereas the subapical network appears unaffected. Scale bar=5 µm. **b, c** Ratio of mean phalloidin fluorescence intensities in the centriolar patches (red dashed lines) and in the rest of the cell surface (white minus red dashed lines) at the apical (**b**) and subapical (**c**) levels. Error bars represent the sem of: at the apical level, $n = 150$ cells for WT, 85 for *Kif3A*-cKO and 103 for *hydin* mutants; 25 for shCtrl, 32 for shLrrc6, 53 for PBS- and 122 for NiCl$_2$-treated explants; at the subapical level, $n = 91$ cells for WT, 50 for *Kif3A*-cKO and 51 for *hydin* mutants; 22 for shCtrl, 32 for shLrrc6,50 for PBS- and 55 for NiCl$_2$-treated explants; from three independent experiments; $P$-values were determined by one-way ANOVA followed by Dunn's multiple comparison test; ***$P \leq 0.0001$; ns, not significant $P > 0.05$. **d** Representative kymographs (200 ms) of motile cilia in brain slices from control (WT) or *hydin* mutant mice or an explant from a WT ventricular wall treated for 12 h with 1 mM NiCl$_2$ or of motile cilia of ependymal cells in culture transfected with shCtrl or shLrrc6. The graph shows the cilia beating frequency (Hz). Error bars represent the sem of $n = 40$ cells for WT and *hydin* mutants; $n = 54$ cells for NiCl$_2$-treated explants, $n = 62$ cilia for shCtrl; $n = 47$ cilia for shLrrc6; from three independent experiments; $P$-values were determined with the Mann–Whitney test; ***$P \leq 0.0001$

mostly expressed in mammalian brain tissue[47,48] and in ciliated epithelial cells in zebrafish[49]. We first verified that Cobl is expressed in differentiating ependymal cells by semi-quantitative RT-PCR in different phases of ependymal cell differentiation in culture[36,37]. Increased expression of Cobl correlated with an increase in the number of multiciliated cells (Supplementary Figure 3a). In the absence of an antibody against mouse Cobl, we localised the Cobl protein in ependymal cells by electroporating the lateral ventricular walls in new-born mice[37,38] with a Cobl-DsRed construct and determined its location at P15. F-actin labelling in mature ependymal cells showed that the Cobl-DsRed is expressed in the apical but not subapical actin network (Fig. 5a).

We then examined, whether Cobl depletion disrupts the apical actin network during ependymal cell development by electroporating an shRNA against mouse Cobl in the brain ventricles at birth. Two shRNA were designed and validated (Supplementary Figure 3b–d, and see below). In immature ependymal cells (P6), no defects in actin organisation or centriole formation, migration or docking were observed in Cobl-depleted cells, showing that Cobl is not needed in early stages of multiciliated ependymal cell development (Supplementary Figure 3e–i). In mature ependymal cells, however, Cobl-depletion significantly decreased actin levels relative to control shRNAs (Fig. 5b–d), and more highly in the apical compared to subapical level. Cobl is, therefore, required for the maintenance of the apical actin network in mature ependymal

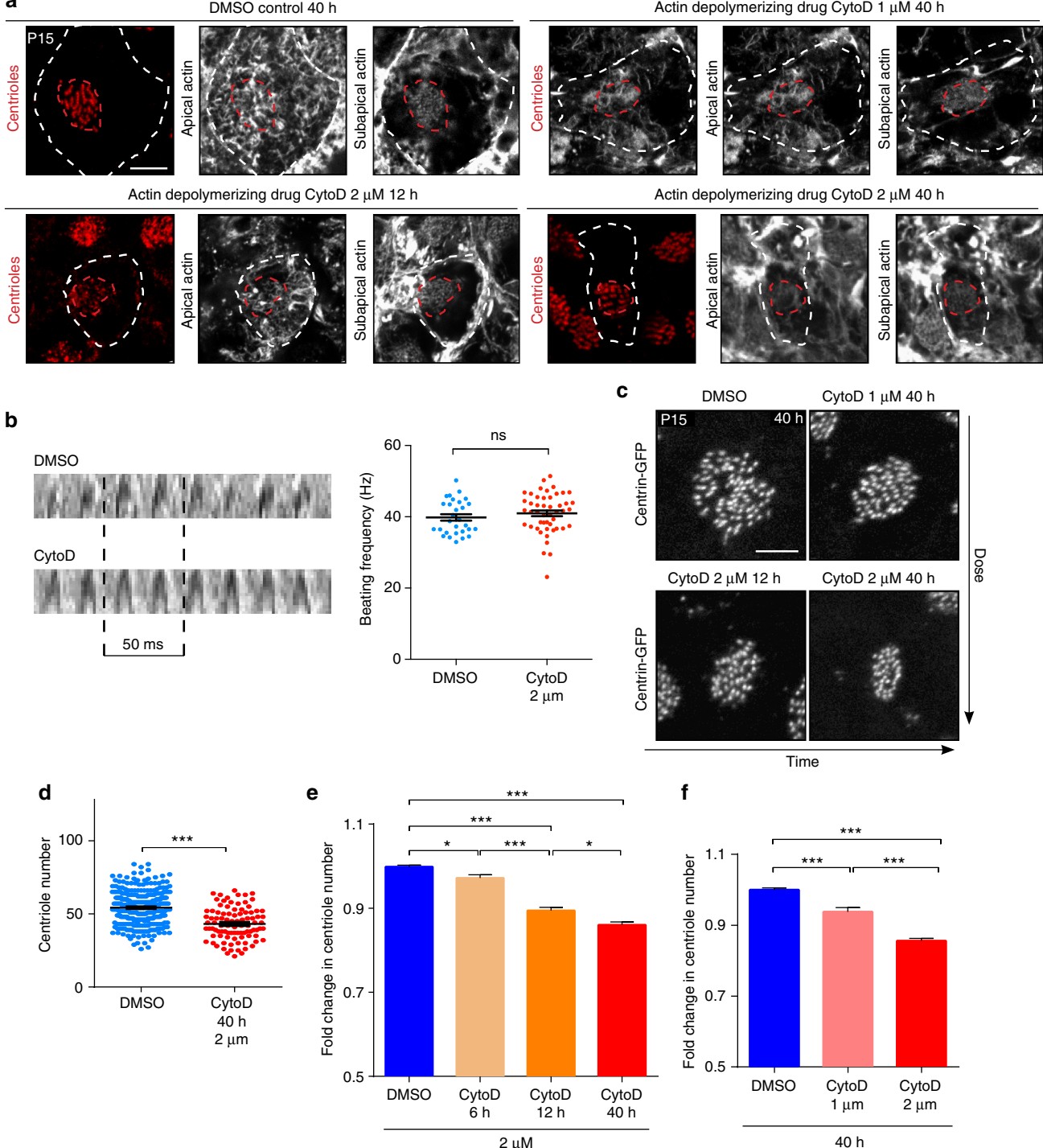

cells. Inter-centriolar distance was unaffected by Cobl depletion (Supplementary Figure 4a), consistent with its localisation at the apical level, confirming that centriolar spacing mainly depends on the subapical actin network. Ependymal cilia beating was still efficient after Cobl depletion, even though the motile cilia were slightly shorter (Supplementary Figure 4b, c), in agreement with a previous report[49]. Indeed, centriole orientation, which is known to depend on cilia beating[37,50], was not randomised in Cobl-depleted cells in vivo (Supplementary Figure 4d, e). Similarly, cilia still beat at a high frequency in Cobl-depleted cells in vitro, only slightly lower than in controls probably due to their shorter length (Supplementary Figure 4f). Remarkably, centriolar patch size and centriole number were significantly decreased in Cobl-depleted cells (Fig. 5e, f; Supplementary Figure 4g). All centrioles were located in the apical plasma membrane stained with the membrane-Cherry electroporation reporter, suggesting that no centrioles were released into the cytoplasm (Fig. 5g). To note, the ciliary tuft size is affected to the same extent that the centriolar patch size by Cobl-depletion (Supplementary Figure 4b, g), suggesting that cilia are lost with centrioles when actin is impaired.

These results show that both Cobl and formins contribute to the enrichment of apical actin at the centriolar patch and the maintenance of centrioles.

**The apical actin protects cilia/centrioles from detachment.** Although, apical actin may be required for centriole maintenance, we noted that centriole loss did not occur in ciliary mutants (either *Kif3A-cKO* or *hydin*) (Supplementary Figure 5a, b) even if actin enrichment is decreased (Fig. 2a, b). We thus hypothesised that the forces exerted by cilia beating may be responsible for the loss of centrioles in the absence of apical actin. We used two different strategies to rescue centriole loss in the absence of apical actin. We first depolymerised actin in P15 explants with cyto-chalasin-D, alone or in association with NiCl2 that blocks cilia beating. Actin was affected by cytochalasin-D treatment in the presence as in absence of NiCl2 (Supplementary Figure 5c). However, centriole loss was prevented if both drugs were combined (Fig. 6a, b). For the second strategy, we depleted Cobl in *Kif3A-cKO* cells, devoid of motile cilia. This depletion induced a decrease of apical actin (Supplementary Figure 5d, e) without affecting centriole number (compare the number of centrioles after Cobl depletion in control Fig. 5e, f or in *Kif3A*-cKO Fig. 6c, d). Thus, the apical actin is required to protect centrioles only when motile cilia are fully active, suggesting that the forces exerted by the movement of cilia and/or the external fluid flow leads to centriole loss when the apical actin is disrupted.

Since no fragmented or whole centrioles were detected inside the cells when the apical actin network was impaired, we hypothesised that the centrioles might have been expelled from cells. To test this hypothesis, we collected the supernatant of mature ependymal cells in culture incubated with DMSO (controls) or cytochalasin-D, alone or in association with NiCl2. Centrioles with their associated cilia were frequently observed in the supernatants of cytochalasin-D-treated cells compared to the DMSO-treated controls or cells in which cilia motility was inhibited by NiCl2 (Fig. 6e, f). This suggests that when the actin network is destabilised, the forces generated by cilia beating cause the centrioles to detach from the apical membrane. To assess whether the flow induced by cilia beating is sufficient to expelled centriole/cilium from the ependymal cells, we applied an artificial external fluid flow[37], comparable to physiological flow, on cells treated with cytochalasin-D and NiCl2. Since the number of centriole/cilium found in the supernatant was not significantly increased compared to controls, it is likely that the shear stress induced by the movement of the cilium itself promotes centriole destabilization in ependymal cells. Nevertheless, we cannot exclude a weak contribution of the collective flow on the expulsion.

When apical actin is decreased (either with drugs or in Cobl-depleted cells) cilia seem to be lost, as well as centrioles (Supplementary Figs. 2a, 4b). No centriole or cilium are observed within the cell. The presence of centriole/cilium in the supernatant of ependymal cells treated with cytochalasin-D suggest that centrioles are expulsed from the cell through the shear stress generated by cilia beating forces, when actin is impaired. Alternatively, drug-induced actin depolymerisation might lead to a centriole/cilium detachment from the cell, while physiologically actin protects centrioles against destabilisation through other mechanisms. Altogether, our results show that cilia motility induces the assembly of a dense apical actin network around centrioles that, in turn, contributes to centriole stabilisation against forces developed by cilia beating/fluid flow (Fig. 6g).

## Discussion

We have demonstrated that, during the maturation of ependymal cells in the developing brain, actin assembles around centrioles at the base of cilia in two functionally different apical and subapical networks. Enrichment of apical actin around the centrioles is induced mechanically by cilia beating. Cilia beating produces different forces that counteract (i) the force of CSF flow ($F_{flow}$) and (ii) the additional torque induced by cilia movement ($F_{torque}$). To determine an order of magnitude for these forces, we simplified the system by considering cilia as cylinders with the geometrical parameters of motile cilia (measured on electron micrographs or on CD24 immunofluorescence), such as radius, $r = 99\,nm \pm 2$ (mean ± sem), centriole length, $L_c = 510\,nm \pm 2$ (mean ± sem) and cilia length, $L = 11.1\,\mu m \pm 0.1$ (mean ± sem)

**Fig. 3** Actin is required for centriole stability in mature ependymal cells. **a** F-actin (phalloidin, grey) and centrioles (FOP, red) in mature ependymal cells from explants treated with DMSO or 1 or 2 μM cytochalasin-D (CytoD) for 12 or 40 h showing concentration- and time-dependent actin impairment; cell borders and centriolar patches are outlined with dashed white and red lines, respectively. **b** Representative 200 ms kymographs of motile cilia in brain slices treated for 40 h with DMSO or 2 μM cytochalasin-D. The graph displays the ciliary beating frequency (Hz) that was not affected by the drug. Error bars represent the sem of n = 48 cells for DMSO- and CytoD-treated explants, from three independent experiments; P-values were determined with the Mann-Whitney test; ns, P > 0.05. **c** Representative images of Centrin2-GFP+ centriolar patches in mature ependymal cells treated with DMSO or 1 or 2μM cytochalasin-D for 12 to 40 h. **d** Representative quantification of the number of centrioles per cell in lateral wall explant at P15 with contralateral brain of the same animal treated with DMSO or cytochalasin-D at 1 μM for 40 h. Error bars represent the sem of n = 301 cells for DMSO- and n = 93 for CytoD-treated explants, from one representative animal out of 6; P-values were determined with the Mann–Whitney test; ***P ≤ 0.0001. **e–f** Fold change in centriole number relative to DMSO-treated contralateral brain in lateral ventricles treated with 1 or 2 μM cytochalasin-D for 6 up to 40 h. A time- (**e**) and dose- (**f**) dependent reduction in the number of centrioles was observed following actin depolymerisation; P-values were determined by one-way ANOVA followed by Dunn's multiple comparison test. Error bars represent the sem of for the dose dependent experiments, n = 389, 388 and 940 cells for DMSO-, CytoD 1μM- and CytoD 2μM-treated explants, respectively, from at least three independent experiments. For the time-dependent experiments, n = 2852, 704, 587 and 889 cells for DMSO-, CytoD 6h-, CytoD 12h- and CytoD 40 h-treated explants from at least three independent experiments. *** P ≤ 0.0001. * DMSO vs. CytoD 6 h P = 0.0287. *CytoD 12 h vs. CytoD 40 h, P = 0.0484. Scale bars=5 μm

(Supplementary Figure 6). These cylinders are considered to be immersed in a medium of low viscosity with their base embedded in a visco-elastic medium simulating the actin network. According to a previous calculation[51], the force counteracting the hydrodynamic flow may be expressed as $F_{\mathrm{flow}} = CfAL$, where $L$ is the length of the cilia, $f$ the beating frequency (Fig. 2d), $A$ the beating amplitude (Fig. 2e) and $C$ is the normal force coefficient

($C = \frac{4\pi\mu}{\mathrm{Log}(L/r)} = 6\times10^{-3}$, where $\mu = 10^{-3}$ Newton s m$^{-2}$ is the viscosity of the CSF[52], and $r$ is the radius of the cilium). Thus, the force developed by a beating cilium at its base to counteract the fluid flow is on the order of $F_{\mathrm{flow}} = 33$pNewton (pN). Cilia beating also induces an additional torque, i.e., a shear stress on the centriole. In response to this torque a higher force, local and temporary, is generated at the base of cilium that can be estimated

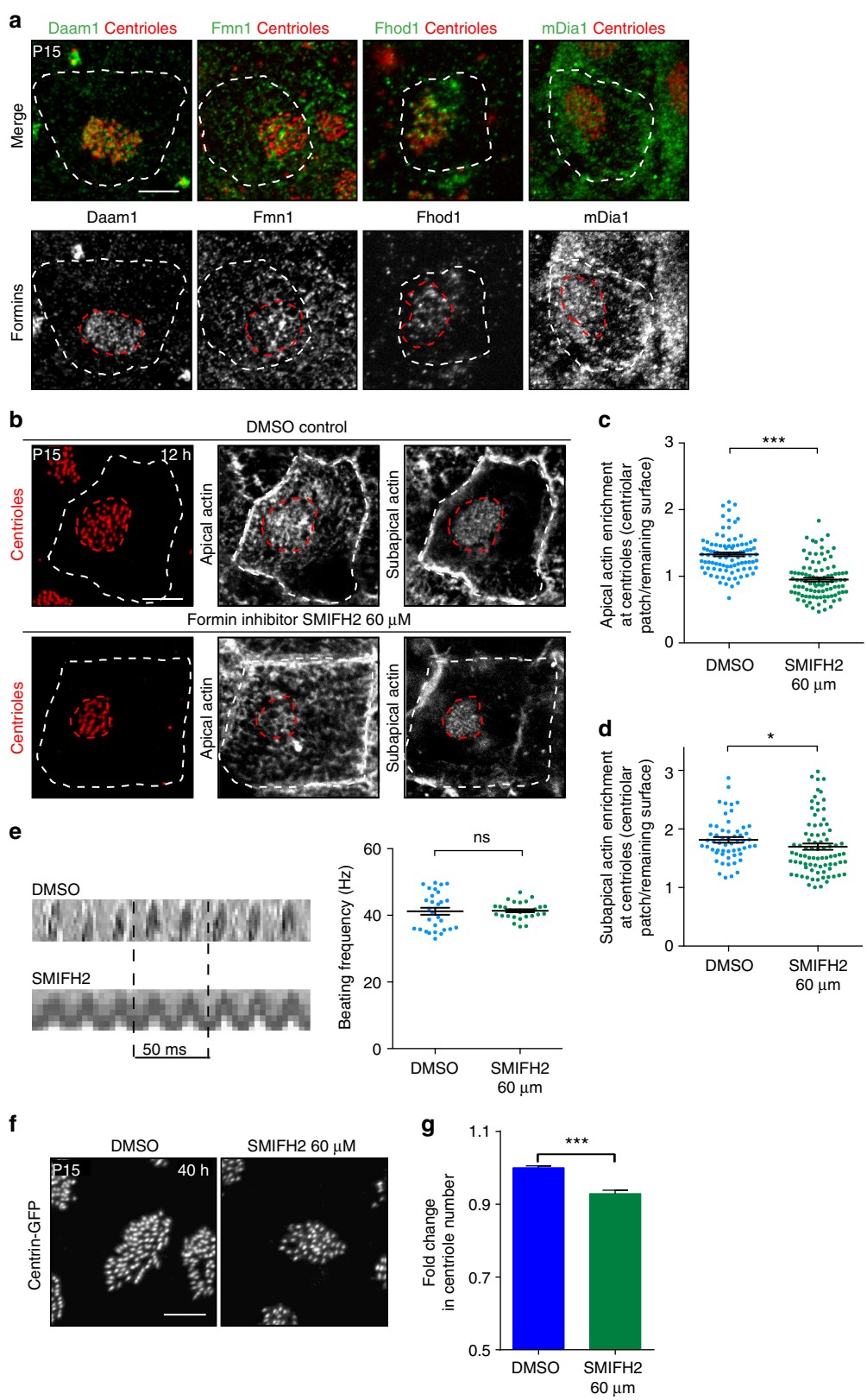

from the equation $F_{torque} = F_{flow}L/L_c$, in which $L_c$ is the size of the centriole stabilised by the actin-rich network. Thus, $F_{torque}$ is on the order of 728pN. Interestingly, this force is similar to those developed by some flagella for propulsion[53]. We have already determined that some formins (mDia1, Fmn1, Daam1 and Fhod1) and WH2-dependent actin nucleator (Cobl) are implicated in the assembly of the apical network. The contribution of formins is particularly noteworthy. Indeed, some members of the formin family, such as mDia1, have been shown to be mechanosensitive[54–57]. Moreover, formin inhibition (using SMIFH2) led to a specific decrease of the apical actin enrichment at the centriolar patch reminiscent of the actin phenotype observed in ciliary mutant or after chemical inhibition of ciliary motility. Further studies are needed to definitely demonstrate that formins are indeed activated by ciliary beating in multiciliated cells.

In turn, the apical actin network protects centrioles against the shear stress, generated by the fluid flow and/or by mechanical torsion. Our in vitro data using drug to induce actin disruption suggest that actin protects centriole by preventing them to be pull out from the cell by motile cilia. However, we cannot rule out other more direct role of actin for protecting centrioles. The direct molecular actors linking centrioles to actin have not been investigated in this study. In this respect, it would be of interest to analyse the role of centriolar actors also involved in the maintenance of centrioles, such as Poc1, FOP or Cep135[11,12,58]. The feedback loop between cilia beating and the structural organisation of the centriolar patch controlled by actin, preserves the appropriate number of motile cilia in ependymal cells and thus maintains efficient CSF flow in the brain ventricles. This maintenance is essential as the CSF contains many signalling molecules that actively regulate mammalian brain development, behaviour (e.g. sleep, appetite) and responses to injury[5]. Moreover, impairment of the CSF flow has also been correlated with age-related dementia[59] and hydrocephalus.

Subapical actin might also help maintain the centrioles, since it was slightly decreased in most experiments. A close relationship between centrioles and the subapical actin network has been noted previously[43], suggesting that the reduction of the enrichment of the subapical actin network observed in our experiments might be due mostly to the decreased number of centrioles. The subapical actin network may be required for the correct spacing of centrioles, as in multiciliated *Xenopus* epidermal cells[14], although the reduction in the inter-centriolar distance might also be influenced by cell shrinkage. Indeed, the inter-centriolar distance was affected by cytochalasin-D that disrupts both the apical and subapical actin networks, but not after SMIFH2-treatment or Cobl depletion that mainly disrupt apical actin enrichment at the centriolar patch.

A role of actin in the maintenance of centrioles has not been observed thus far in other multiciliated tissues or organisms[14,60]. This could be due to insufficient depolymerisation of the apical actin network in previous studies, since we have shown that the disappearance of centrioles depends on the dose and duration of treatment with the actin-depolymerising agent.

Many questions remain to be answered such as: What is the detailed structural organisation of the apical and subapical actin networks? What nucleators initiate the subapical network? What roles are played by each of the various formins localised at the apical region of centrioles? Are these formins mechanically activated by cilia beating? What specific roles are played by the formins and Cobl, or what signalling pathways activate them? Interestingly, the role of RhoA-activated actin assembly by Fmn1 in the emergence of the apical surface of multiciliated cells has recently been reported[17,18].

## Methods

**Transgenic mice**. All animal studies were performed in accordance with the guidelines of the European Community and French Ministry of Agriculture and were approved by the Ethic comity Charles Darwin (C2EA-05) and "Direction départementale de la protection des populations de Paris", (Approval number Ce5/2012/107; APAFiS #9343). The mouse strains have already been described: RjOrl: SWISS (Janvier labs), Cen2–GFP[61] (CB6-Tg(CAG-EGFP/CETN2)3-4Jgg/J, The Jackson Laboratory); B6-$Kif3a^{tm2Gsn}$ ; B6-Ift88$^{tm1Bky}$[30], C57BL/6-Tg(hGFAP-Cre)[62], C57BL/6-Tg(Nestin-Cre)[63], *hydin* (B6CBACa Aw-J/A-Hydin hy3/J, The Jackson Laboratory[64]). To produce hydin homozygous mutants, we crossed hydin$^{+/-}$ or hydin$^{+/-}$/ Cen2-GFP with hydin$^{+/-}$ or hydin$^{+/-}$/ Cen2-GFP. To produce the conditional mutant *Kif3a* (*Kif3A*-cKO), we crossed $Kif3a^{fl/fl}$ or $Kif3a^{fl/fl}$/ Cen2–GFP mice with $Kif3a^{ko/+}$/ Tg(hGFAP-Cre) mice. To produce the conditional mutant *Ift88* (*Ift88* cKO), we crossed $Ift88^{fl/fl}$ mice with $Ift88^{ko/+}$/ Tg(Nestin-Cre). Wild-type littermate mice were used as controls.

**Plasmids**. Cobl, derived from the mouse Cobl complementary DNA clone mKIAA0633 (Kazusa DNA Research Institute, Japan), was cloned in pDsRed-N1 (Clontech). To determine siRNA sequences to obtain Cobl depletion, we screened ten siRNA sequences against Cobl-GFP ectopically expressed in Hela cells. Two were selected (Supplementary Figure 3b): Cobl sh1: 5′ACCGGTGCATGGTTC-TAGTCACATTCAAGAGATGTGACTAGAACCATGCACTTTTTCTCGAGG3′ and Cobl sh2: 5′ACCGGTCATCGACTCAAGACTATTTCAAGAGAA-TAGTCTTGAGTCGATGACTTTTTCTCGAGG3′. We also selected an shRNA corresponding to an Lrrc6 shRNA previously designed on human sequences from a homologous region[32]: 5′ACCGGTGCCCAAGGTAGGAGAAATGATTTCAA-GAGAATCATTTCTCCTACCTTGGGCACTTTTTCTCGAGG3′. We cloned the shRNA sequences in the shRNA backbone vectors pcDNA-U6-shRNA and pTRIP-U6-shRNA-IRES-PGK-GFP-WPRE, as previously described[65]. The control shRNA (shCtrl) was generated from AllStars Negative Control siRNA (Qiagen), which does not recognise mammalian RNAs. The reporter plasmid encoding membrane Cherry (mCherry) under the control of the CAGGS promoter was co-transfected with pcDNA-U6-shRNA and was a gift from Xavier Morin. All cells expressing the shRNA-GFP (pTRIP-U6-shRNA-IRES-PGK-GFP-WPRE) or cells expressing pCAGGS-mCherry, which were co-transfected with the untagged pcDNA-U6-shRNA, were considered to be depleted.

**Postnatal electroporation**. Electroporation of plasmids encoding sh1Cobl or sh2Cobl or shLrrc6 or shCtrl (U6 backbone untagged) (1.25 μg) and the reporter

**Fig. 4** Apical actin nucleated by formins contributes to centriole stability in mature ependymal cells. **a** Mature (P15) ependymal cell centrioles (labelled with FOP or Cep164; red) and formins (green and grey); note that all formins colocalise with the centriolar patch (red dashed lines); cell borders are outlined with dashed white lines. **b–d** Lateral wall explants from P15 mice treated for 12 h with DMSO or SMIFH2. **b** F-actin (phalloidin, grey) and centrioles (FOP, red) in mature ependymal cells from DMSO- or SMIFH2-treated explants. Cell borders and centriolar patches are outlined with dashed white and red lines, respectively. In control (DMSO-treated) explants, apical actin is enriched in the centriolar patch; formin inhibition prevents enrichment. Subapical actin staining is slightly affected by SMIFH2-treatment. **c, d** Ratio between mean phalloidin fluorescence intensities in the centriolar patch (red dashed lines) and in the remaining cell surface (white minus red dashed lines) at the apical (**c**) and subapical (**d**) levels. Error bars represent the sem of: at the apical level, $n = 96$ cells for DMSO- and SMIFH2-treated explants; at the subapical level, $n = 56$ and 82 cells for DMSO- and SMIFH2-treated explants, respectively, from three independent experiments. $P$-values were determined by the Mann–Whitney test; *$P = 0.0156$; ***$P \leq 0.0001$. Scale bars=5 μm. **e** Representative kymographs (200 ms) of motile cilia in P15 brain slices treated for 12 h with DMSO or SMIFH2 (formin inhibitor). The graph shows the cilia beating frequency (Hz); formin inhibition had no effect. Error bars represent the sem of $n = 29$, 48 cells for DMSO- and SMIFH2-treated explants from three independent experiments; $P$-values were determined with the Mann–Whitney test; ns, $P > 0.05$. **f** Representative images of Centrin2-GFP$^+$ centriolar patches. **g** Fold change in centriole number relative to DMSO-treated contralateral brain in lateral ventricles treated with SMIFH2, showing a reduction of the centriole number after SMIFH2-treatment; $P$-values were determined with the Mann–Whitney test. Error bars represent the sem of $n = 1572$ cells for DMSO- and 466 for SMIFH2-treated explants, from three independent experiments; ***$P \leq 0.0001$. Scale bars=5 μm

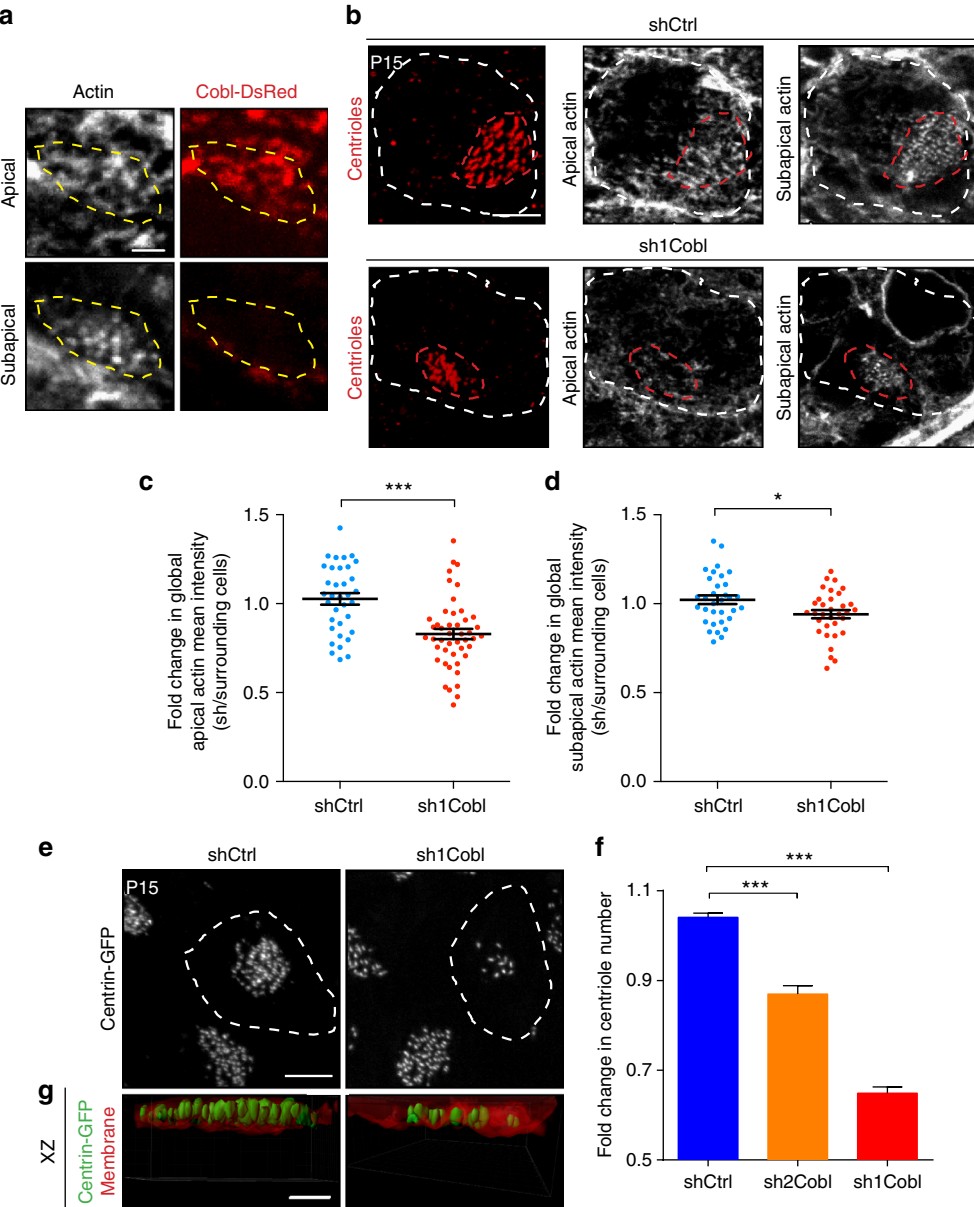

**Fig. 5** Apical actin nucleated by Cobl also contributes to centriole stability in mature ependymal cells. **a** F-actin (phalloidin, grey) and Cobl (Cobl-dsRed, red) in P15 lateral ventricular walls of wild-type mice electroporated at birth with the COBL-dsRed construct; COBL-dsRed is specifically localised in the apical but not subapical actin networks. (yellow dashed lines = centriolar patch). **b** F-actin (phalloidin, grey) and centrioles (FOP, red) in mature ependymal cells in P15 wild-type mice electroporated at birth with shCtrl or sh1Cobl and the mCherry reporter. Cell borders of transfected cells and centriolar patches are outlined with dashed white and red lines, respectively. **c**, **d** Mean phalloidin fluorescence intensities inside the border of transfected cells relative to the mean of the three closest non-transfected control cells at the apical (**c**) and subapical (**d**) levels; apical and to a lesser extent subapical actin networks decrease in Cobl-depleted cells. Error bars represent the sem of: at the apical level, $n = 35$ cells for shCtrl and 46 for sh1Cobl transfected cells; at the subapical level, $n = 33$ cells for shCtrl and sh1Cobl from three independent experiments. $P$-values were determined by the Mann–Whitney test; $*P = 0.0390$, $***P \leq 0.0001$. **e–g** Mature ependymal cells on lateral ventricular walls of P15 Centrin2-GFP mice transfected at birth with shCtrl or sh1Cobl or sh2Cobl (dashed white line). **e** Representative images of Centrin2-GFP$^+$ centriolar patches. **f** Fold change in centriole number in the sh-transfected cells (shCtrl, sh1 or sh2Cobl) relative to those in surrounding cells. Error bars represent the sem of $n = 236$ cells for shCtrl, 129 for sh2Cobl and 106 for sh1Cobl transfected cells from three independent experiments $P$-values were determined by the Mann–Whitney test; $***P \leq 0.0001$. **g** 3D-modelling of XZ projections of centrioles (green) docked in the plasma membrane (mCherry reporter, red). Scale bars=5 μm in **b**, **e** and 1 μm in **a**, **g**

plasmid mCherry (0.75 μg) was performed on neonates to two-day-old pups, as previously described[38]. This co-transfection was used throughout the paper to deplete Cobl except when co-transfected with the plasmid encoding Cobl-dsRed where sh1Cobl or shCtrl in the pTRIP backbone (1.25 μg) were used.

**Immunostaining.** Lateral walls of the lateral brain ventricles were dissected, fixed, and immunostained, as previously described[66]. The same area of the lateral walls was used throughout the study (Fig. 1a). The tissue was treated with 0.1% triton in

BRB80 (80 mM K-Pipes pH6.8; 1 mM MgCl2; 1 mM Na-EGTA) for 1 min and fixed in methanol at −20 °C for 10 min except for phalloidin stained tissues, which were detergent-treated and then fixed in 4% PFA in BRB80 for 8 min. The following antibodies were used: mouse IgG1 GT335 (1:2000, Adipogen AG-20B-0020-C100); rat anti-CD24 (1:200, BD Biosciences 557436); mouse IgG1 GTU88 (1:500, Sigma T6557); rabbit anti-Cep164 (1:500, NOVUS 45330002); mouse IGg2b anti-AKAP450 (1:20, CTR453 gift from M. Bornens); rat anti-rootletin (1:50, Santa Cruz sc-67824); mouse IgG2b anti-Centrins 20H5 (1:2000; Millipore 04-1624); rabbit or rat anti-dsRed (1:200, Clontech 632496 or 5F8 1:400, Chromotek 5f8-

100); chicken anti-GFP (1:800, Aves GFP-1020); rabbit anti-Daam1 (1:50, Euromedex 14876-1-AP); rabbit anti-Fhod1 (1:75, Euromedex FP3481); mouse anti-Fmn1 (1:100, Abcam AB68058); rabbit anti-mDia1 (1:200, BD Biosciences 610848); mouse IgG2b acetylated-tubulin 6-11b1 (1:600, Sigma T6793); mouse IgG2b anti-FOP (FGFR10P, 1:500, Abnova H00011116-MO) and species-specific Alexa Fluor secondary antibodies (1:800, Invitrogen). Phalloidin-A488 was used at a dilution of 1:50 with primary and secondary antibodies; Phalloidin-A568 was used at a dilution of 1:40 for the STED experiment.

**Optical imagery**. Images (230-nm or 210-nm z-steps) of fixed samples were obtained with either an inverted epifluorescence microscope (Zeiss Axio Observer Z1) equipped with a Plan-Apochromat ×63 (NA 1.4 oil-immersion) objective, a Zeiss Apotome with an H/D grid, a CCD camera (ORCA-ER, Hammamatsu), and Zen2 software, or a confocal microscope (model SP5 or SP8; Leica) with a ×63 (Plan Neofluar NA 1.3 oil-immersion) objective and Leica LAS-AF software. Z-stack projections along the entire apical surface or 6 z-stacks encompassing the apical or subapical actin networks (apical, subapical actin) are shown.

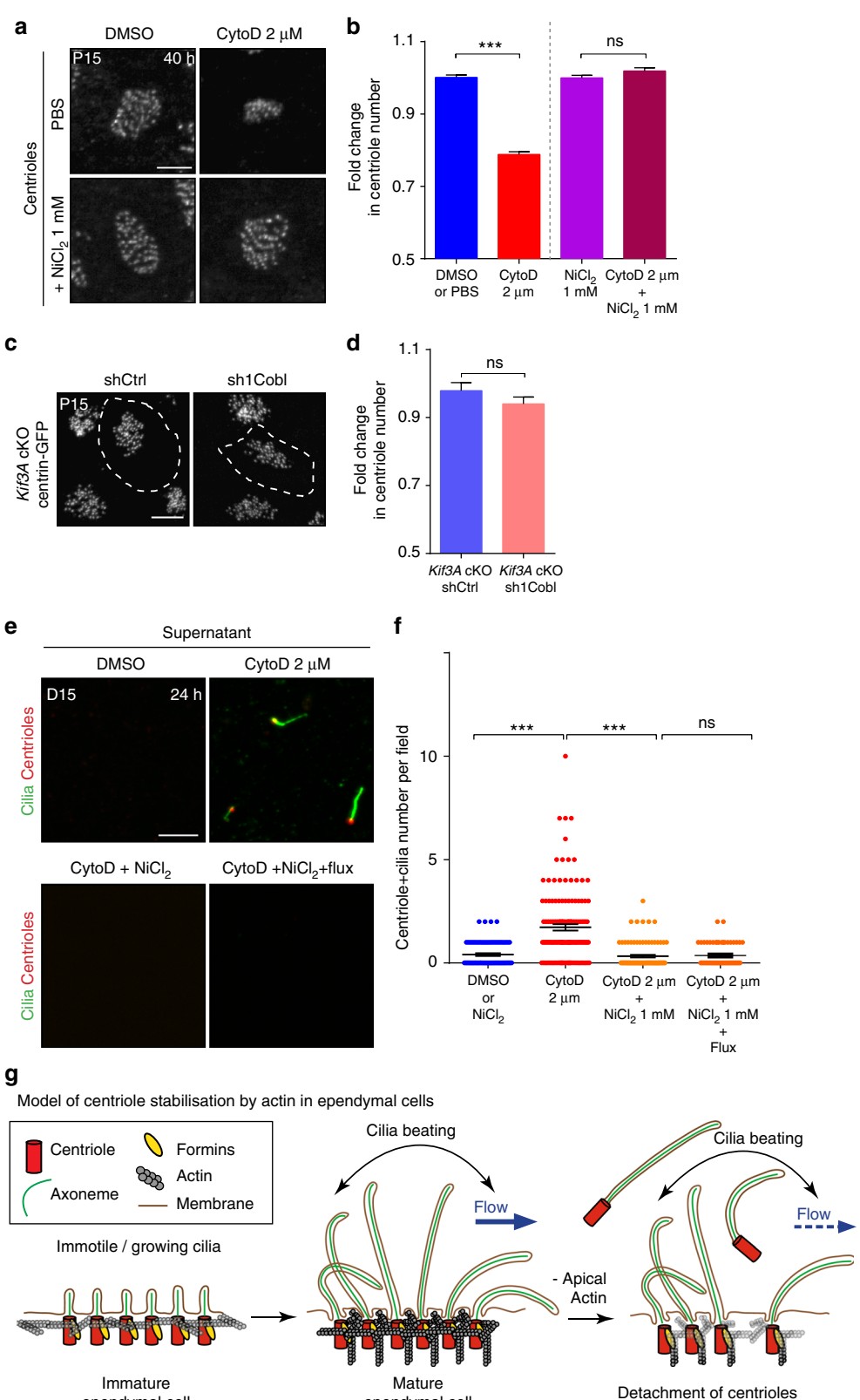

**STED nanoscopy**. Images were acquired by gCW STED microscopy (TCS SP8-3×; Leica Microsystems) with a ×100 oil immersion objective (NA 1.4 oil-immersion); parameters were optimised for Alexa 532, Alexa 568, and Alexa 488 detection. Samples (pixel size = 30 nm) were excited sequentially at 575 nm, 530 nm, and 490 nm wavelengths with a supercontinuum laser; a 660 nm depletion laser was used. For Alexa 568, a 20% AOTF conventional scanner (400 Hz, Line Average 2, Accumulation 3) and a 60% depletion laser were used; fluorescence (585–610 nm) was collected with a hybrid detector (Gain 30%) in the gated mode (0.2–6 ns) and a 1 Airy Unit pinhole. For Alexa 532, a 20% AOTF conventional scanner (400 Hz, Line Average 2, Accumulation 3) and a 100% depletion laser were used; fluorescence (540–575 nm) was collected using a hybrid detector (Gain 30%) in the gated mode (0.2–6 ns) and a 1 Airy Unit pinhole. For GFP, a 10% AOTF conventional scanner (400 Hz, Line Average 2, Accumulation 2), and a 100% depletion laser were used; fluorescence (500–530 nm) was collected using a hybrid detector (Gain 30%) in the gated mode (0.5–6 ns) and a 1 Airy Unit pinhole. Stack images were acquired in 0.4 µm Z-steps.

**Phase-contrast video-microscopy**. Cilia beating frequencies were analysed with an inverted epifluorescence microscope (Leica, DMIL LED) equipped with an apochromat 63× (NA 1.32 oil-immersion) objective, a FASTCAM mini AX50 camera (Photron) and Photron fastcam software. Mature ependymal ciliary beatings were recorded at 250 images per second in phase contrast with transmitted light.

**Transmission electron microscopy**. Transmission electron microscopy was performed, as previously described[37]. Brains from P21 mice were sectioned into 200 µm slices using a vibratome (VT 1200S, Leica). Sections of the rostral lateral and dorsal walls of the lateral ventricle were taken apart in PBS under a dissecting microscope. Ventricle slices were fixed in 2.5% glutaraldehyde and 4% paraformaldehyde. Tissues were treated with 1% OsO4. Fixed specimens were washed and progressively dehydrated. Samples were then incubated in 1% uranyl acetate in 70% methanol. Final dehydratations were made. Samples were pre-impregnated with ethanol/epon mix (2/1, 1/1, 1/2 ratios subsequently), and impregnated into epon. Samples were then mounted into epon blocks for 48 h at 60 °C to ensure polymerisation. Ultrathin sections of 70 nm were cut using an ultramicrotome (Ultracut E, Reichert-Jung) and analysed using a Jeol 10–11 transmission electron microscope.

**Brain slices**. Freshly harvested P15 brains were cut in Hank's solution (1 × HBSS, 10 mM Hepes, 0.075% sodium bicarbonate, 1% Penicillin/Streptomycin) in 200 µm coronal sections on a vibratome (Leica VT1200S) and placed in Dulbecco's Modified Eagle's-Glutamax Medium (DMEM-Glutamax, Invitrogen) supplemented with 10% FCS, 1% penicillin/streptomycin.

**Drug treatments**. Lateral wall explants or brain slices were immersed for the indicated times at 37 °C with 5% $CO_2$ in DMEM-Glutamax (Invitrogen) supplemented with 10% FCS, 1% penicillin/streptomycin and cytochalasin-D to depolymerise actin, SMIFH2 to inhibit formin[42] or $NiCl_2$ to block cilia beating[39] (Sigma). The lateral ventricular wall of the opposite side of each mouse brain was used as control and immersed in the same media containing an amount of DMSO or PBS equivalent to that of the drug. Phalloidin fluorescence after a 12 h or 40 h incubation shows early and late defects in the actin cytoskeleton.

**Cell cultures**. Differentiating ependymal cells isolated from mouse brain were cultured, as previously described[66]. The shRNA constructs (0.06 µg) and the Tomato reporter vector (0.04 µg) were co-transfected into $0.25 \times 10^6$ cells with the Jet Prime kit (polyplus transfection) when the cells were harvested for differentiation, according to the manufacturer's instructions.

**Supernatant collection**. Ependymal cells, derived from Centrin2-GFP mice and differentiated in culture for 13–15 days, were treated for 24 h with 2 µM cytochalasin-D or DMSO with or without 1 mM $NiCl_2$ in DMEM-Glutamax (Invitrogen) supplemented with 10% FCS, 1% penicillin/streptomycin. To generate a flow comparable to physiological values for mature cells we used the rotating unit described previously[37], with the rotating plates at a height of 4 mm and a rotation speed of 6 rounds/min. Clockwise and anticlockwise rotations were used equally in our experiments.

The supernatant was then collected and Taxol (2 µM) was added to stabilise cilia. The supernatant was then centrifuged for 10 min at 150 g to remove cell debris and 30 min at $12000 \times g$ to concentrate centrioles. The pellet was resuspended in 4 ml of DMEM-Glutamax supplemented with 1% penicillin/streptomycin and 2 µM taxol and sedimented on coverslips coated with Poly-L-lysine by centrifugation at $10,400 \times g$ for 10 min at 4 °C. The cells were fixed in cold methanol for 10 min and the cilia stained with the GT335 antibody in PBS/0.1% Tween/3% BSA.

**RT-PCR**. Total RNA was extracted from ependymal cells with the RNeasy Midi Kit (Qiagen), according to the manufacturer's instructions, at different time points during differentiation in vitro. RNA was converted into single-stranded cDNA using the Superscript-III Reverse Transcription System (ThermoFisher) with random primers, according to the manufacturer's instructions. The following primers were used for subsequent PCR amplification: the housekeeping gene CyclophilinA Fw primer 5′-ACCCCACCGTGTTCTTCGAC-3′ and Rev primer 5′-CATTTGC CATGGACAAGATG-3′ for normalisation; the Cobl Fw primer 5′-TGAGATCC AAGGACAAATGG-3′ and Rev primer 5′-CTCATCTCTGATTTGGGAGG-3′. CyclophilinA and Cobl were amplified simultaneously by PCR in the same tube and the products subjected to agarose gel electrophoresis. Band intensities were quantified with ImageJ software and normalised with respect to cyclophilinA.

**3D-modelling**. Imaris software was used for 3D-modelling.

**Data analysis**. All analyses were performed on images of the rostro-dorsal part of the lateral walls of the lateral ventricles to maximise reproducibility (Fig. 1a).

**Measurement of phalloidin fluorescence intensity**. We compared actin enrichment (phalloidin fluorescence) in the centriolar patch (delineated by centriole staining) of mature (P15) ependymal cells in ciliary mutant mice, explants treated with SMIFH2 or DMSO and Lrrc6-depleted cells. The mean intensity of phalloidin fluorescence was measured in the centriolar patch and on the surface of the rest of the cell; cell borders and actin linking the centriolar patch to the cell membrane were excluded. Apical actin was analysed on the sum projection of the 6 z-stacks encompassing Cep164 staining and subapical actin of the 6 z-stacks encompassing Akap450 staining with ImageJ software. The results are expressed as the ratio of the values in the centriolar patch over the remaining cell surface.

Total mean actin (phalloidin fluorescence) levels were measured with ImageJ software in the apical and subapical networks (as defined above) at early (P4 and P6 mice) or late (P15 mice) stages of differentiation in ependymal cells transfected with sh1Cobl or shLrrc6 or shCtrl within cell borders, in the sum projection of transfected cells normalised with respect to the mean signal intensity in the three closest non-transfected ependymal cells at the same stage, using ImageJ software. The results are expressed as the ratio of the values of the transfected cells over the mean values of the three surrounding control cells.

**Fig. 6** Forces generated by cilia beating cause centrioles/cilia to detach from actin-deficient ependymal cells. **a**, **b** Mature (P15) ependymal cells from explants treated for 40 h with PBS or DMSO (Control), 1 mM $NiCl_2$ to block cilia beating, 2 µM cytochalasin-D (cytoD) to depolymerise actin, or with both drugs; **a** γ-tubulin stained centrioles. **b** Fold change in centriole number relative to DMSO- or $NiCl_2$-treated contralateral brain, showing that cilia beating induces centriole destabilisation in absence of actin; P-values were determined by one-way ANOVA followed by Dunn's multiple comparison test. n = 781 cells for control-, 597 for CytoD-, 819 for $NiCl_2$- and 547 for CytoD + $NiCl_2$-treated explants, from three independent experiments; ***$P \leq 0.0001$; ns $P > 0.05$. **c**, **d** Lateral walls of Centrin2-GFP ciliary mutant mice (Kif3A cKO) electroporated at birth with shCtrl or sh1Cobl and a reporter construct (represented as white dashed lines) and fixed at P15. **c** Representative Centrin2-GFP$^+$ centriolar patches. **d**, Fold change in centriole number relative to surrounding cells, showing no effect of Cobl-depletion on centriole number in ciliary mutant mice. P-values were determined by the Mann–Whitney test. ns, $P > 0.05$. n = 101 electroporated cells for Kif3A-cKO shCtrl and Kif3A-cKO sh1Cobl from three independent experiments. **e** Representative images of the supernatant of mature ependymal cells from Centrin2-GFP mice in culture. After 15 days of the onset of differentiation, cells were treated for 24 h with cytochalasin-D or DMSO with or without 1 mM $NiCl_2$ and an external fluid flow; isolated centrioles (Centrin2-GFP, red) with a cilium (GT335, green) were observed in the supernatant of CytoD-treated cultures. **f** Number of centrioles associated with a cilium per 63 × field. n = 76 fields for DMSO-, 112 for CytoD-, 71 for CytoD + NiCl2- and 61 for CytoD + $NiCl_2$ + Flux-treated cell cultures, from three independent experiments; P-values were determined with the Mann–Whitney test; ***$P \leq 0.0001$. Scale bars=5 µm. **g** Working model: Ependymal cilia beating leads to actin enrichment at the centriolar patch. Apical actin is in turn crucial for stabilising centrioles exposed to cilia beating and fluid flow forces. In cells with impaired apical actin, beating cilia cause centrioles and their associated cilia to detach from the cells. Error bars represent the sem in all graphs

**Quantification of cilia length**. Cilia length was measured on whole-mount preparations of P15 lateral wall and stained with GT335 or 6-11b1 or CD24 that recognise glutamylated-tubulin, acetylated-tubulin and axonemal membrane, respectively. The length was evaluated by measuring the distance between the most peripheral centrioles at the periphery of the patch and the longest stained cilia in the same area. Three measurements per cells were averaged. The different markers labelled different portions of cilia. As different markers and fixation were used, the length was normalised with respect to DMSO-treated explants or surrounding non-transfected cells (control).

**Quantification of centriole number**. Centriole numbers were determined on methanol-fixed Centrin2-GFP mice or WT samples stained with γ-tubulin, as indicated. The distribution of the number of centrioles in mature ependymal cells varied between animals from different litters (different sizes) of different genetic backgrounds. We thus normalised centriole number in shRNA transfected cells with respect to non-transfected surrounding cells. The distribution of the number of centrioles is equivalent in the two hemi-brains of a given animal. Thus, in drug-treated explants, centriole number was normalised against the number of centrioles in the opposite lateral ventricular wall treated with PBS or DMSO. The distribution of the number of centrioles was also equivalent in two animals from the same litter. Thus, centriole numbers in ciliary mutants were normalised with respect to the mean number of centrioles in littermate controls.

In the shRNA experiments, centrioles, represented by Centrin2-GFP dots in 0.23 μm z-sections to include all centrioles of each cell, were counted manually. The pilot experiment, performed blindly, was similar to subsequent experiments.

Centrin2-GFP dots in the pharmacological experiments were counted automatically with an in-house ImageJ plugin, using 3D-Gaussian and 3D-LoG filter[67] blurring to remove noise and artefacts in a 3D-volume. Local 3D-maxima were then extracted from the filtered image with Fast Filters 3D[68] to determine the centre of the centrins in 3D-space. The parameters of the filters are based on average 3D centriole size and are constant in all conditions. Corrections were made manually. The centriolar patch boundaries were estimated on the basis of the spatial density of the centrins and corrected manually. The pilot experiment was also quantified manually, blind to the experimental conditions; the results were equivalent to those obtained in the experiments counted automatically.

**Quantification of centriolar patch size**. We also determined, in each cell, the size of the centriolar patch, defined as the area of the minimum convex xy euclidean space that contains all the centrioles in the cell. Centriolar patch size, on all cells, was calculated with Matlab software. Patch size was very sensitive to the presence of scattered centrioles, because all of the centrioles were taken into consideration.

**Distance to the nearest neighbour**. For each patch or cell, we also determined the nearest neighbour xy euclidean distance between centrioles. These distances were extracted using the cell profiler plugin.

**Cilia beating frequency and amplitude**. Three square 16 pixels$^2$ regions of interest (ROI), each containing a region with a single beating cilium, were selected in the transfected cells in culture (12 to 14 days after serum withdrawal) or in mature ependymal cells in brain slices (P14–15 animals). The number of beats (frequency in Hz) was counted with the ImageJ z-axis profile and presented as a scattered dot plot. Kymographs were obtained for the same regions using the Multiple Kymograph ImageJ plugin of J. Rietdorf and A. Seitz. Amplitude was determined on the same high-speed recording by measuring the distance between the maximal and minimal position of the tip of individual cilia.

**Determination of centriole orientation**. The in-house ImageJ plugin (see above) was used to detect centrin2-GFP dots and Cep164 staining with manual corrections. Inside each cell, centrins were assigned to their corresponding Cep164 (unit vector) using the Hungarian algorithm[69] that minimises the sum of the distances between all pairs of points. The angle between the unit vectors and their mean direction were calculated. The centriolar patch boundaries were estimated from the spatial density of the centrins and corrected manually. Circular plots with a 20° binning interval were obtained from the resulting angles. The in-house Matlab toolbox for generating vector plots and the corresponding analyses can be downloaded from https://github.com/biocompibens/centriole_analysis.git.

**Statistics**. Means of two conditions were compared with the Mann–Whitney test. Means of more than two conditions were analysed by one-way ANOVA followed by Dunn's multiple comparison tests (GraphPad Prism 6.0). The Kolmogorov–Smirnov or Waston-U2 tests were used to compare whole angle distributions shown as circular plots (MatLab). The $\chi^2$ test for trends was used to compare centriole neighbour distances (GraphPad Prism 6.0). Scatter plots are presented as the mean±sem. ROUT methods were used to remove outliers generated by plugins analysis of automatic counts, with a Q value of 1% (GraphPad Prism 6.0). All results were obtained in at least three independent experiments on three different animals or cell cultures.

**Code availability**. Codes and plugins can be consulted at https://github.com/biocompibens/centriole_analysis.

**Data availability**. The data that support the findings of this study are available within the article and supplementary files, or available from the authors upon request.

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

## Acknowledgements

We thank the IBENS Imaging Facility, member of France-BioImaging, supported by the French National Research Agency (ANR-10-INBS-04, ANR-10-LABX-54 MEMO LIFE, ANR-11-IDEX-0001-02 PSL) and by the "Région Ile-de-France" (NERF No. 2011-45), the "Fondation pour la Recherche Médicale" (DGE 20111123023) and the "Fédération pour la Recherche sur le Cerveau-Rotary International France" (2011). We thank the "Fondation Imagine" for supporting the acquisition of the Stimulated emission depletion microscope. We are grateful to L. Goldstein, B. Yoder and J. Gleeson for providing the *Kif3a*-cKO, *Ift88*-cKO and centrin2-GFP mice. We thank X. Morin for the CAAGS-mCherry plasmid, M. Bornens for the CTR453/AKAP450 antibody and I. Caillé and S. Scotto for the shRNA backbone. We are grateful to C. Auger, A. Correia, and D. Varela for their help with the animal husbandry. We greatly appreciated the helpful discussions with P. Bastin, A. Benmerah, J.F. Joanny and all the members of the Spassky laboratory. This study was supported by the CNRS, the ENS, the INSERM, the Fondation Pierre-Gilles de Gennes (FPGG03), the Agence Nationale de la Recherche (ANR-12-BSV4-0006 CILIASTEM), La Ligue contre le Cancer- Comité de Paris (RS14/75-88), the program ≪Investissements d'Avenir≫ (ANR-10-LABX-54 MEMO LIFE, ANR-11-IDEX-0001-02 PSL), the Fondation pour la Recherche Médicale (20140329547), the INCA grant 2014-1-PL BIO-11 and the European Research Council (Consolidator grant "EDeN" #647466 to N.S., Senior grant "forcefulactin" #249982 to M.-F.C.).

## Author contributions

A.Mahuzier performed and analysed experiments; A.S., N.M. and A.G. designed the ImageJ, Cell Profiler and Matlab plugins used in this study; C.F. designed the siRNA and the Cobl construct; P.L. performed electroporations; M.F. optimised electroporation experiments; A.Meunier performed the electron microscopy used to measure the centriole diameter and length; M.G.-T. developed the STED microscopy; R.V. calculated the force developed at the base of beating cilia; M.-F.C. initiated the study; N.S. designed the study and supervised the project; N.D. initiated the study, designed, performed and analysed experiments and supervised the project; N.D., N.S., and A.Mahuzier wrote the manuscript. All authors commented on the manuscript.

## Additional information

**Competing interests:** The authors declare no competing interests.

