## [Peer Review File · Nature Communications]

REVIEWERS' COMMENTS:

Reviewer #1 (Remarks to the Author):

Overall the manuscript is improved, although I still think cell size could be significantly driving centriole number. Since the only data actually dealing with cilia loss is done with drug treatments it is hard to correlate this with centriole number used throughout the paper. This is a limitation of the paper.

Comments:

More description of the analysis of ciliary length is needed. I find it difficult to believe the length of cilia in a patch can be accurately scored with the described methods. This concern is increased by the "representative" image that shows cilia appear clearly shorter in S2A (CytoD). Related minor point... It would also appear in Fig S2A based on the size of centrioles in the various panels that the scale bar does not accurately reflect all images. This should be addressed.

The "modeling" is poorly developed, poorly rationalized and adds nothing of substance to the paper. This kind of modeling does not belong in a data driven paper of this quality. The fact that the authors include it in the discussion rather than the results highlights the lack of real data driving this model.

Reviewer #3 (Remarks to the Author):

The authors have addressed my concerns satisfactorily.

Reviewer #1 (Remarks to the Author):

Overall the manuscript is improved, although I still think cell size could be significantly driving centriole number.

Cell size could indeed be significant in driving centriole number. Actually, we observed a correlation between cell size and the number of centrioles within cells in untreated conditions. However, if most actin disruption led to a perturbation of cell size, depleting Cobl led to actin disruption without changing the cell size. In this condition, centrioles are destabilised without affecting cell size, whereas other actin disruptions (using Cytochalasin-D or SMIFH2) led to both decrease of the number of centriole and decrease of the cell size. Thus, the change in cell size is not a pre-requisite for centriole destabilisation.

Since the only data actually dealing with cilia loss is done with drug treatments it is hard to correlate this with centriole number used throughout the paper. This is a limitation of the paper.

It is visible on the supplementary figures S2a and S4b that the number of cilia is decreased as well as the number of centrioles in both drug-driven and Cobl-depletion-driven actin disruption. This suggest that centriole loss correlates with cilium loss whether we used drugs or not. However, it was indeed too challenging to quantify the number of cilia themselves within in a ciliary tuft. We thus toned down our conclusion to say that “the presence of centriole/ cilium in the supernatant of ependymal cells treated with cytochalasin-D suggest that centrioles are expelled from the cell, through the shear stress generated by cilia beating forces, when actin is impaired. Alternatively, drug-induced actin depolymerisation might induce a centriole/ cilium detachment from the cell, while physiologically actin protects centrioles against destabilisation through other mechanisms”.

Comments:

More description of the analysis of ciliary length is needed. I find it difficult to believe the length of cilia in a patch can be accurately scored with the described methods.

We now provide a more detailed description of our methods to analyse ciliary length.

This concern is increased by the “representative” image that shows cilia appear clearly shorter in S2A (CytoD). Related minor point... It would also appear in Fig S2A based on the size of centrioles in the various panels that the scale bar does not accurately reflect all images. This should be addressed.

Thank you for pinpointing this mistake. Actually the images for the Cytochalasin-D panel were acquired on tissue fixed with paraformaldehyde whereas the other images were from cells fixed with Methanol. These different fixations led to different rendering in centriole and cilia size. That was another reason to normalise our measurement to counterpart brain treated at the same time and in the same conditions than the actual sample. We now replace the image with sample all fixed in Methanol and hope the images will appear more convincing.

The “modelling” is poorly developed, poorly rationalized and adds nothing of substance to the paper. This kind of modeling does not belong in a data driven paper of this quality. The fact that the authors include it in the discussion rather than the results highlights the lack of real data driving this model.

We understand the concerns of reviewer 1 for the physical model, as this is not that usual in data-driven paper. We now added the real data driving the calculation on the Supplementary Figure S6. We strongly believe that the model brings an extra value to the paper by giving an order of idea of the force developed by beating cilia. This might be useful for complementary studies on the activation of formin through forces. We thus would like to keep the model in the discussion but are ready to remove it if the editor finds it necessary.

Reviewer #3 (Remarks to the Author):

The authors have addressed my concerns satisfactorily.